# Atmospheric wind energization of ocean weather

Shikhar Rai [1,2], J. Thomas Farrar [2] & Hussein Aluie [1,3,4] ✉

Ocean weather comprises vortical and straining mesoscale motions, which play fundamentally different roles in the ocean circulation and climate system. Vorticity determines the movement of major ocean currents and gyres. Strain contributes to frontogenesis and the deformation of water masses, driving much of the mixing and vertical transport in the upper ocean. While recent studies have shown that interactions with the atmosphere damp the ocean's mesoscale vortices $O(100)$ km in size, the effect of winds on straining motions remains unexplored. Here, we derive a theory for wind work on the ocean's vorticity and strain. Using satellite and model data, we discover that wind damps strain and vorticity at an equal rate globally, and unveil striking asymmetries based on their polarity. Subtropical winds damp oceanic cyclones and energize anticyclones outside strong current regions, while subpolar winds have the opposite effect. A similar pattern emerges for oceanic strain, where subtropical convergent flow is damped along the west-equatorward east-poleward direction and energized along the east-equatorward west-poleward direction. These findings reveal energy pathways through which the atmosphere shapes ocean weather.

Atmospheric wind is the primary energy source maintaining the global ocean circulation[1]. While wind provides kinetic energy (KE) to the ocean at gyrescales >$10^3$ km, most of the ocean's KE resides at the "mesoscales" $O(100)$ km[2]. Mesoscales are the ocean's weather systems[3], consisting of a seemingly amorphous tangle of vortical and straining motions[4]. There is strong evidence that in fact winds have a net damping effect on the mesoscales—a process sometimes called "eddy-killing"[5–10]. The process substantially modifies the KE input to the ocean[6,10–13], can weaken the ocean gyres[14], and feeds back onto the gyrescale currents such as the Gulf Stream[15] and Antarctic Circumpolar Current (ACC)[16].

To explain eddy-killing, previous studies[6–8,17] focused primarily on wind stress $\boldsymbol{\tau}$ being proportional to wind velocity $\mathbf{u}_a$ relative to the ocean velocity $\mathbf{u}$, which induces a curl in wind stress, $\nabla \times \boldsymbol{\tau}$, with polarity opposite to the ocean mesoscale vorticity $\nabla \times \mathbf{u}$, resulting in net negative wind work. However, there remains a fundamental gap in understanding how wind work on the ocean is related to the ocean's

vortical and straining motions. While it is possible to derive budgets for ocean vorticity and strain, these do not provide the energy channeled into vortical and straining motions by wind stress. An occasional misconception is that a Helmholtz decomposition can separate vorticity from strain, with the latter mistakenly regarded as being solely due to the potential flow accounting for divergent motions (see Supplementary Information, SI). In fact, strain is also an essential constituent of divergence-free (or solenoidal) flows, including the oceanic mesoscales in geostrophic balance where strain-dominated regions account for approximately half the KE (Figs. S5 and S6). There is no existing fluid dynamics framework that relates the injection of KE by a force to how this energy is deposited into vortical and straining motions. This is fundamental to our understanding of eddies and how they evolve, and to our ocean weather forecasting capabilities[18,19].

Below, we show that winds, on average, are just as effective at damping straining motions as they are at damping vortical motions. This happens because oceanic strain induces a straining wind stress

[1]Department of Mechanical Engineering, University of Rochester, Rochester, NY, USA. [2]Department of Physical Oceanography, Woods Hole Oceanographic Institution, Woods Hole, MA, USA. [3]Department of Mathematics, University of Rochester, Rochester, NY, USA. [4]Laboratory for Laser Energetics, University of Rochester, Rochester, NY, USA. ✉e-mail: hussein@rochester.edu

gradient (WSG), which is analogous to ocean vorticity inducing a curl in wind stress. Ocean-induced WSGs alone, whether straining or vortical, always damp ocean currents. However, our theory also reveals that a significant contribution to wind work comes from inherent wind gradients, a main component of which is due to prevailing winds of the general atmospheric circulation. We find that inherent WSGs lead to asymmetric energization of ocean weather based on the polarity of vortical and straining ocean flows.

## Results

### Scale physics of wind work
We probe the scale physics of air-sea momentum exchange using a coarse-graining methodology, which yields a first-principles measure of power input by wind stress $\boldsymbol{\tau}$ into the surface ocean currents $\mathbf{u}$ at all length-scales $< \ell$[10],

$$EP_\ell = \overline{(\boldsymbol{\tau} \cdot \mathbf{u})}_\ell - \overline{\boldsymbol{\tau}}_\ell \cdot \overline{\mathbf{u}}_\ell. \tag{1}$$

Here, $EP_\ell$ stands for eddy power input by winds into all oceanic motions smaller than $\ell$[10] and $\overline{(\ldots)}_\ell$ denotes spatial filtering on the sphere[20]. More details are in "Methods" (see also refs. [10,21]). By probing different length-scales $\ell$ in Eq. (1), it was recently shown[10] that wind work at scales smaller than $\ell = 260$ km is negative on a global average due to eddy-killing, which has a clear seasonal cycle and is concentrated in regions with high KE such as the Gulf Stream, Kuroshio, and the ACC. Here, we shall disentangle the contribution to $EP_\ell$ from vortical and straining oceanic flow patterns and also from the global atmospheric circulation patterns.

**Leading-order approximation.** We accomplish this through an approximation of $EP_\ell$ in Eq. (1) that is rooted in a systematic multiscale expansion due to Eyink[22]. The approximation relies on the so-called "ultraviolet scale-locality" of wind work[23–25]. Ultraviolet locality of a multiscale process such as $EP_\ell$ is a fundamental physical property indicating that contributions from scales $\delta < \ell$ to the process at scale $\ell$ decay at least as fast as a power-law of the scale disparity ratio $\delta/\ell$ (see "Methods" and ref. [26]). For $EP_\ell$, ultraviolet locality is valid over scales $\ell \approx 100$ km and smaller, but not over scales larger than the mesoscale spectral peak at $\approx 300$ km[2]. The leading order term, $\widetilde{EP}_\ell$, in the multiscale expansion yields an approximation of $EP_\ell$,

$$EP_\ell \approx \widetilde{EP}_\ell = \frac{1}{2} M_2\, \ell^2\, \boldsymbol{\nabla}\overline{\boldsymbol{\tau}}_\ell : \boldsymbol{\nabla}\overline{\mathbf{u}}_\ell, \tag{2}$$

with $M_2 = 0.4$ (see "Methods" for derivation). We shall demonstrate below the utility of $\widetilde{EP}_\ell$ for disentangling air-sea momentum exchanges due to various flow patterns. First, we present evidence that that $\widetilde{EP}_\ell$ is indeed an accurate proxy for $EP_\ell$ using data from both satellites (Fig. 1) and a high-resolution coupled general circulation model (Fig. S1). In Fig. 1 (and Fig. S1), panels A and B show global maps of $EP_\ell$ and $\widetilde{EP}_\ell$, respectively, time-averaged over the dataset record. As expected, $\widetilde{EP}_\ell$ provides a good approximation to $EP_\ell$, with a correlation coefficient of 0.9. Panels C and D in Fig. 1 (and Fig. S1) show instantaneous maps of $EP_\ell$ and $\widetilde{EP}_\ell$ in the Kuroshio Extension region on Dec 13, 1999 (Fig. 1) and in the Gulf Stream region on day 10 of the first year of the model output (Fig. S1). These maps also show good spatial agreement between $EP_\ell$ and its approximation $\widetilde{EP}_\ell$ with a correlation coefficient of 0.9. Agreement between $EP_\ell$ with $\widetilde{EP}_\ell$ also holds in time: their time-series are compared in panel E of Fig. 1 (and Fig. S1), in the Gulf Stream and ACC regions, which exhibit the pronounced seasonality found in ref. [10]. Furthermore, panel F of Fig. 1 (and Fig. S1), shows a global map of the correlation coefficient between the time-series of $EP_\ell$ and $\widetilde{EP}_\ell$ at every geographic location; the average of the correlation coefficient is very high ($\geq 0.9$) with a relatively small standard deviation ($\approx 0.05$).

### Energization of strain and vorticity
Having established that $\widetilde{EP}_\ell$ can accurately approximate the full expression for wind work on the ocean's mesoscales and sub-mesoscales, we are now able to disentangle contributions from vortical and straining oceanic flow patterns. The wind work proxy can be decomposed exactly as

$$\widetilde{EP}_\ell = \underbrace{\frac{1}{2} M_2 \ell^2 \overline{\mathbf{T}}_\ell^{strn} : \overline{\mathbf{S}}_\ell}_{\widetilde{EP}^{strn}} + \underbrace{\frac{1}{2} M_2 \ell^2 (\boldsymbol{\nabla} \times \overline{\boldsymbol{\tau}}_\ell) \cdot \overline{\boldsymbol{\omega}}_\ell}_{\widetilde{EP}^{vort}}. \tag{3}$$

Here, $\widetilde{EP}^{strn}$ and $\widetilde{EP}^{vort}$ are wind work on the ocean's mesoscale straining ($\overline{\mathbf{S}}_\ell = (\boldsymbol{\nabla}\overline{\mathbf{u}}_\ell + \boldsymbol{\nabla}\overline{\mathbf{u}}_\ell^{\mathrm{tr}})/2$) and vortical ($\overline{\boldsymbol{\omega}}_\ell = \boldsymbol{\nabla} \times \overline{\mathbf{u}}_\ell$) motions, respectively, where superscript '$(\ldots)^{\mathrm{tr}}$' indicates the tensor transpose. See "Methods" for more details. Equation (3) shows that the ocean's vortical motions are coupled only to the curl of the wind stress, whereas the ocean's straining motions are coupled to the straining component of wind stress, $\overline{\mathbf{T}}_\ell^{strn} = (\boldsymbol{\nabla}\overline{\boldsymbol{\tau}}_\ell + \boldsymbol{\nabla}\overline{\boldsymbol{\tau}}_\ell^{\mathrm{tr}})/2$.

Figure 2 provides example snapshots of $\widetilde{EP}^{vort}$ and $\widetilde{EP}^{strn}$ in panels A and B, respectively. Both panels visualize the same streamlines of the ocean currents in a region in the north Pacific. It can be seen in Fig. 2A that $\widetilde{EP}^{vort}$ is concentrated in vortical regions occupied by eddies, while Fig. 2B shows how $\widetilde{EP}^{strn}$ is concentrated in strain-dominated regions outside eddies where the flow is an amorphous tangle. These panels demonstrate how our theory and the resultant relation (3) successfully decompose wind work into straining and vortical contributions. Such disentanglement cannot be accomplished properly using traditional eddy detection methods such as the Okubo-Weiss criterion[27], which is a binary designation of a geographic location as either strain-dominated or vorticity-dominated even though strain and vorticity are often collocated (see Fig. S2) and can lead to severe errors in energy transfer estimates (by more than 6×, see Fig. S3A versus Fig. S4A). Figure 2A reveals a striking asymmetry whereby cyclonic (anti-clockwise in the northern hemisphere, NH) vortices are damped by wind (blue, $\widetilde{EP}^{vort} < 0$), whereas anticyclonic (clockwise in NH) vortices are energized by wind (red, $\widetilde{EP}^{vort} > 0$). Figure 2B shows an analogous effect of wind on straining motions. We now provide an explanation for this phenomenon.

### Asymmetric energization of ocean weather
Recent studies[6,8,10,13] have highlighted the importance of wind stress variations induced by the ocean's mesoscale vortical motions. These induced wind stress gradients (WSGs) due to the ocean's vortical eddies are often sketched as in Fig. 2C1, which illustrates how the dependence of wind stress on wind velocity relative to the ocean velocity leads to a damping of a vortical ocean eddy even when the wind velocity itself is uniform and lacks any gradient[6,8]. This mechanism for eddy-damping is captured by $\widetilde{EP}^{vort}$ in Eq. (3), where part of $\boldsymbol{\nabla} \times \overline{\boldsymbol{\tau}}_\ell$ arises due to induced WSGs.

However, past studies have overlooked induced WSGs due to the ocean's mesoscale straining motions. Their role is naturally revealed by $\widetilde{EP}^{strn}$ in Eq. (3), which we sketch in Fig. 2D. Figure 2D1 illustrates how the ocean's straining motions (green), when coupled to background winds of uniform velocity, induce a straining wind stress (Fig. 2D2) that opposes the ocean's strain.

Our analysis provides a complete theory for how ocean current-induced WSGs always act to oppose the oceanic mesoscale flow, both vortical and straining motions, as sketched in Fig. 2C, D. The previously unrecognized mesoscale damping of strain is significant and accounts for approximately half of the mesoscale damping by wind shown in Fig. 1B (see Figs. S5 and S6). The other half is due to the mesoscale damping of vortical motions[6,8].

In addition to induced WSGs, our theory accounts for the effect of inherent wind gradients on oceanic mesoscales. Most previous studies

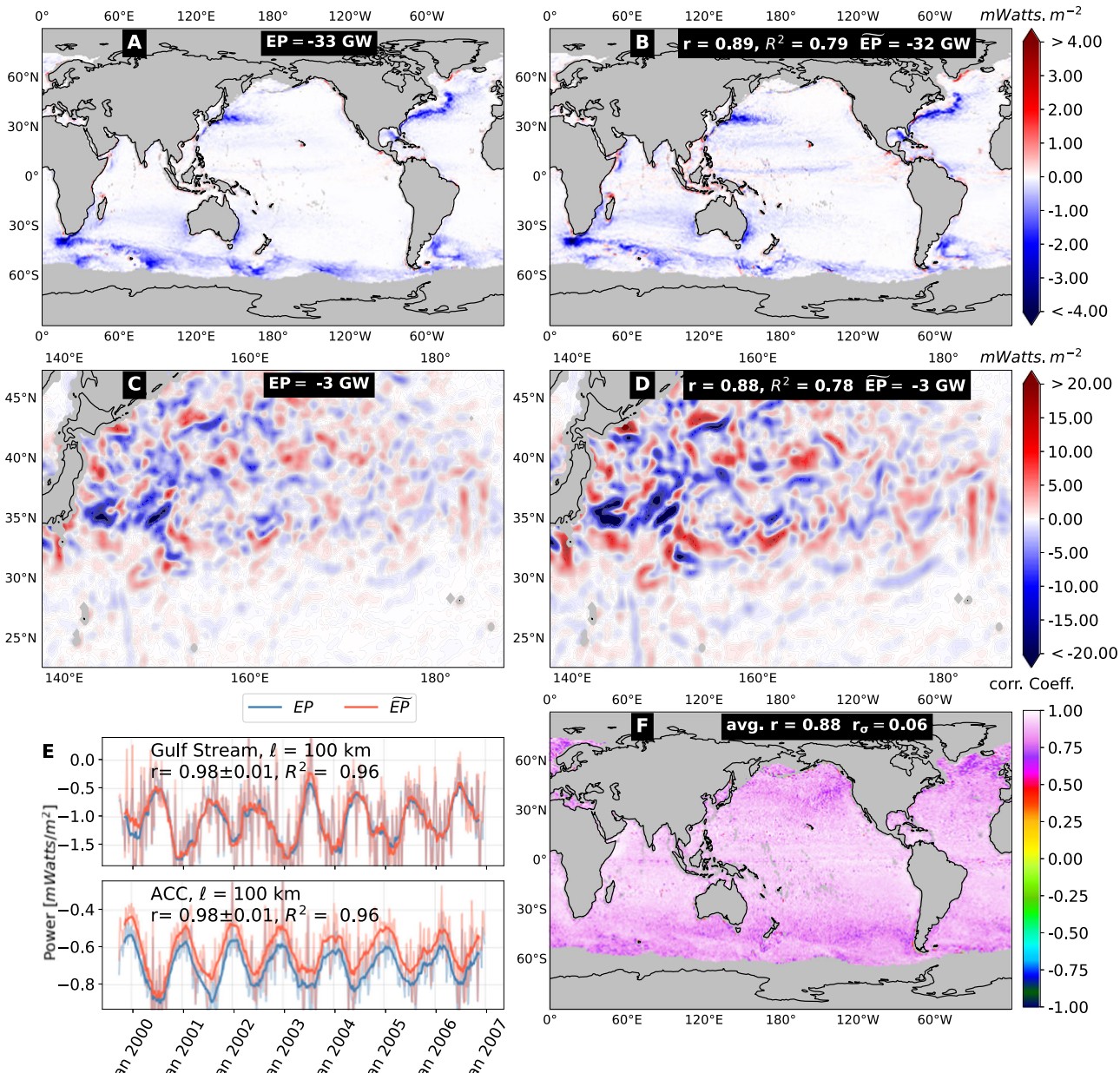

**Fig. 1 | Measuring wind energization of ocean weather via $\widetilde{EP}$, which is an excellent proxy for wind power input, $EP$, and affords us insight into the roles of mesoscale vortical and straining motions comprising ocean weather.** **A**, **B** Compare $EP$ and $\widetilde{EP}$ (averaged from Oct 1999 to Dec 2006) at $\ell$ = 100 km using satellite data. **C** and **D** compare $EP$ and $\widetilde{EP}$ on a single day (Dec 13, 1999) in the Kuroshio Extension region. These panels demonstrate that $EP$ in (**A**, **C**) can be accurately approximated by $\widetilde{EP}$ in (**B**, **D**), which display a high Pearson correlation coefficient $r \approx 0.9$ and coefficient of determination $R^2 \approx 0.9$. **A–D** Display the area-integrated values of $EP$ or $\widetilde{EP}$ (in Giga Watts, GW). **E** Time series of $EP$ and $\widetilde{EP}$ in the Gulf Stream and in the ACC, which again exhibit high correlation ($r = 0.98$) and demonstrates that $\widetilde{EP}$ captures the seasonality of $EP$ reported recently[10,49]. **F** A global map of the temporal correlation between $EP$ and $\widetilde{EP}$ at every location, and shows that $r = 0.88 \pm 0.06$, which reinforces our treatment of $\widetilde{EP}$ as an accurate proxy for $EP$.

analyzed wind work on the mesoscales using a Reynolds decomposition[11,13,15,28], which does not incorporate the role of inherent wind gradients. Those studies focused on analyzing the quantity $\langle \boldsymbol{\tau}' \cdot \mathbf{u}' \rangle$, where $\langle \dots \rangle$ represents a temporal or spatial average and $(\dots)'$ represents fluctuations about that average. In contrast, the quantity $EP_\ell$ in Eq. (1), which arises from the coarse-graining framework, naturally accounts for the role of wind gradients that are inherent to the global atmospheric circulation patterns. This role of inherent WSGs appears in the $\boldsymbol{\nabla}\overline{\boldsymbol{\tau}}_\ell$ factor of $\widetilde{EP}_\ell$ in Eq. (2).

Figure 2E, F explains how a spatially varying stress due to the prevailing winds acts on mesoscales and submesoscales. Panel E decomposes wind stress into a uniform (translational) stress and a

shear stress. The latter can be further decomposed into vortical and straining wind stress components as shown in Fig. 2E. Panel F sketches the orientation of vortical and straining stresses from inherent gradients of the prevailing zonal (east-west) winds in the subtropics due to the atmospheric planetary circulation. Since these stresses arise from inherent spatial gradients in the prevailing winds, they do not always act to oppose the mesoscale flow, unlike WSGs induced by ocean currents.

In the subtropics, we see from Fig. 2F that the inherent WSG is anticyclonic, i.e., it has negative curl in the northern hemisphere (NH) and positive curl in the southern hemisphere (SH). This imparts an asymmetry to wind work on vortical motions, whereby cyclonic eddies

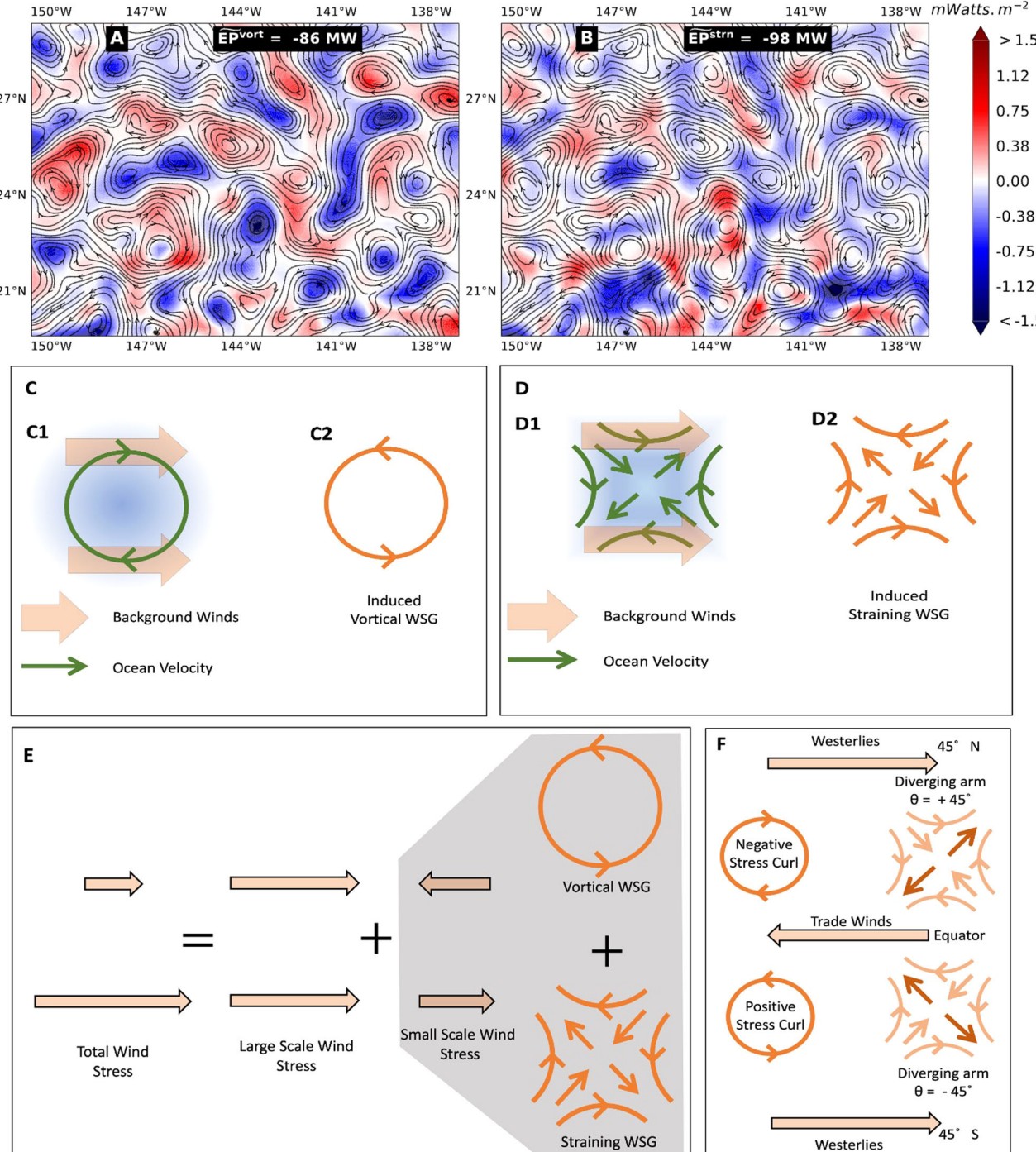

**Fig. 2 | Disentangling ocean weather energization by winds.** Energy transfer (in Mega Watts, MW) from atmospheric winds into ocean (**A**) vorticity, $\widetilde{EP}^{vort}$ and (**B**) strain, $\widetilde{EP}^{strn}$, can be analyzed using our theory and are shown here in a region in the north Pacific on Dec 6, 1999. Streamlines in (**A**) and (**B**) are identical and visualize the ocean surface currents. While vortical and straining motions always co-exist at every location, some regions can be dominated by one or the other. Demonstrating our approach, (**A**) shows that wind work on vorticity is most pronounced inside ocean eddies (closed streamline contours), whereas (**B**) shows that wind work on strain has comparable magnitude but dominates in regions outside eddies where the flow is an amorphous tangle. **C**, **D** Describe the mechanisms of current-induced wind stress gradients (WSGs), which always oppose ocean currents, thereby damping both ocean vorticity and strain. In (C1), when ocean vortical flow **u** (green) encounters uniform background winds $\mathbf{u}_a$, it experiences stress proportional to the relative wind velocity $\mathbf{u}_a - \mathbf{u}$, which induces a curl in wind stress

(vortical WSG) that always opposes the ocean eddy (C2). (C1) differs from the usual sketch of eddy-killing (e.g., refs. 6,10) by scale-decomposing the relative wind stress, as in (**E**), into a sum of spatially uniform stress and smaller-scale shearing stress: the former can only lead to bulk motion but cannot transfer energy to/from ocean vortical or straining flows, which is done by the latter. In (D1), similar to (C1), when ocean strain (green) encounters uniform winds, it induces a straining WSG that always opposes the ocean strain (D2). Ocean-induced WSGs (**C**, **D**) alone always damp ocean currents and do not explain the positive wind work (red) in (**A**) and (**B**). This is explained by inherent wind gradients, which naturally arise in our theory, with an important component being due to the prevailing trade winds and westerlies sketched in (**F**). These lead to asymmetric energization of ocean weather based on the polarity of vortical (anti/cyclonic) and straining (positive/negative angle $\theta$) flows.

are damped and anticyclonic eddies are energized by the subtropical prevailing winds. It explains the wind work patterns seen in Fig. 2A. Panel F in Fig. 2 shows that there is a similar asymmetry in how the prevailing winds force straining mesoscale motions.

We separate straining motions based on the angle $\theta$ made by the strain's diverging arm with the zonal (east-west) direction, which ranges from $-90°$ to $+90°$. In analogy with the curl being either positive or negative, strain can have either $\theta > 0°$ or $\theta < 0°$. Using this partitioning of strain, the subtropical winds in the NH/SH exert $\theta > 0°/\theta < 0°$ straining stress on the oceanic mesoscales.

From the preceding discussion, wind work on the mesoscales is determined by the combination of inherent and induced wind gradients. This is made precise by simplifying Eq. (3) to

$$\widetilde{EP}_\ell \approx \alpha\,|\mathbf{u}_a|\,\ell^2 \left[ \underbrace{\overline{\mathbf{S}}_a : \overline{\mathbf{S}} + \overline{\boldsymbol{\omega}}_a \cdot \overline{\boldsymbol{\omega}}}_{\text{inherent}} - \underbrace{(|\overline{\mathbf{S}}|^2 + |\overline{\boldsymbol{\omega}}|^2)}_{\text{induced}} \right], \qquad (4)$$

where $\alpha = \rho_{air}\, C_D M_2/2$ with air density $\rho_{air}$ and drag coefficient $C_d$ (see "Methods" for details). The induced portion of wind work is negative semi-definite, damping the mesoscales. The inherent portion of wind work depends on the mesoscale flow's orientation relative to wind gradients as described in Fig. 2E, F. In regions where the mesoscales are very strong (see Fig. S5C and S5D), such as in western boundary currents, the induced portion in Eq. (4) is always greater than that due to inherent WSGs and eddy-damping dominates regardless of the mesoscale flow orientation. However, in the remaining 90% of the world's oceans as shown in Fig. 3, inherent WSGs play a central role. That wind work on mesoscales is the outcome of a competition between inherent and induced WSGs was not recognized before, which our theory is able to quantify via Eq. (4).

Figure 3 (also Fig. S7) reveals a markedly asymmetric wind work on mesoscales depending on their polarity. Figure 3A shows wind work on vortical mesoscale motions with positive curl ($\widetilde{EP}_{\omega>0}^{vort}$), which resembles the curl of wind stress ($\nabla \times \overline{\boldsymbol{\tau}}$) in Fig. 3C. Similarly, Fig. 3B shows wind work on vortical mesoscale motions with negative curl ($\widetilde{EP}_{\omega<0}^{vort}$), which is almost the exact opposite of that in panel A. Asymmetric wind energization by the prevailing winds may explain, at least partly, the observed asymmetry of cyclonic versus anticyclonic eddies[29,30], and the fact that anticyclonic eddies have a longer lifetime than cyclonic eddies[31,32].

An analogous effect exists for the mesoscale strain. We see in Fig. 3D that wind work on straining mesoscale motions having a diverging arm with angle $\theta > 0°$ resembles the straining component of wind stress $((\nabla\overline{\boldsymbol{\tau}}_\ell + \nabla\overline{\boldsymbol{\tau}}_\ell^{\text{tr}})/2)$ in Fig. 3F. On the other hand, wind work on straining mesoscale motions with $\theta < 0°$ is almost the exact opposite of that in panel D.

In Fig. 3, we see a contrasting behavior in strongly eddying regions such as in the Gulf Stream, Kuroshio, and ACC. In these regions, the induced contribution dominates in Eq. (4) rendering it negative such that winds damp mesoscale motions regardless of polarity.

Without the decomposition of wind work in Fig. 3 (and Fig. S7) based on polarity of the ocean's mesoscale vorticity and strain, the asymmetric energization by the prevailing winds would not be apparent. Indeed, the sum of panels A, B, D, E in Fig. 3 yields $\widetilde{EP}_\ell$ in Fig. 1B, which is negative almost everywhere in the global ocean. Even a decomposition of wind work on mesoscale strain and vorticity separately (see Fig. S5), without distinguishing polarities, indicates that these oceanic motions are damped on average. This is because the induced WSGs contribution in Eq. (4) is persistently negative regardless of the ocean flow's orientation. When ocean weather of mixed polarity passes through any geographic location (Eulerian perspective), the imprint of inherent winds can seem significantly smaller than what the actual ocean weather system experiences (Lagrangian perspective).

Seasonality of wind work on mesoscale vorticity and strain of each polarity is shown in Fig. 3G, H at subtropical latitudes, excluding regions of strong mesoscale KE (see "Methods"). Magnitudes of wind work peak during winter of each hemisphere, regardless of whether wind acts to energize or dampen the mesoscales. This can be understood from Eq. (4), where $\widetilde{EP}_\ell$ is proportional to the mesoscale strength, which peaks in spring[2], and to wind speed $|\mathbf{u}_a|$, which peaks in winter. Since absolute seasonal variations of the latter are much larger, they govern the seasonality of wind work[10].

Traditional techniques such as eddy detection[32,33] or Okubo-Weiss[27] are poor at revealing the inherent asymmetry of energy transfer we found in Fig. 3 (also Fig. S7). The deficiency is demonstrated in Figs. S8 and Fig. S9, especially from the energization time-series in panels [G] where wind work on anti-cyclonic flow oscillates around zero and lacks a clear seasonal cycle. In contrast, the counterpart time-series in panels [G] of Fig. 3 and Fig. S7 using our theory are clearly positive and with a seasonal cycle. The deficiency is due to vorticity-dominated regions, as detected by these methods, having significant contributions from strain of either sign and vice versa, which contaminate the time-series.

## Discussion

Our theory quantifies wind work on ocean weather, which consists of an amorphous tangle of straining and vortical mesoscale motions. While previous studies[9,10,13,33] focused on estimating the kinetic energy exchange between the atmosphere and the ocean, a fundamental gap remained in understanding how such energy feeds or damps the ocean's vortical and straining flows. Budgets for ocean vorticity and strain do not quantify energy transfer and we have been lacking a fluid dynamics framework that relates the two perspectives. We are able to derive such a relation here by using a coarse-graining approach[10] combined with insights into how disparate scales are coupled (scale-locality)[23,26] and a multiscale expansion[22].

We found that, on average, wind damps oceanic strain at a rate equal to the damping of oceanic vorticity. The damping of strain had not been recognized before and has important implications to the formation of ocean fronts[34,35]. Yet, underlying the damping of strain and vorticity is a marked asymmetry whereby wind energizes ocean weather with certain polarities. Such asymmetry arises because wind work is an outcome of a competition between induced and inherent wind stress gradients. The induced component of wind work is always negative and proportional to oceanic mesoscale strength, dominating in strong current regions, which occupy less than 10% of the ocean surface[10]. Wind work over the remaining 90% of the ocean surface is dominated by the inherent component of wind stress gradients, which energize ocean weather with certain polarities and may explain observed asymmetry of anti/cyclonic systems in the ocean[32]. These results reveal the energy pathways through which the atmosphere shapes ocean weather. We hope that an improved understanding of air-sea energy transfer can be integrated into predictive models. This is especially pertinent to climate models, which are often unable to resolve mesoscales accounting for over 50% of the global oceanic circulation's KE[2].

## Methods
### Description of datasets
Our results are based on two sets of data, one from satellites shown in the main text, and another from a high resolution coupled global model. The satellite dataset is for seven years (Oct 1999 to Dec 2006), which includes geostrophic ocean currents estimated from satellite altimetry (AVISO) and wind stress from QuikSCAT scatterometry, both projected onto a 0.25° × 0.25° grid. Geostrophic currents along the equator are calculated using Lagerloef methodology[36] with the $\beta$ plane approximation. Wind stress is calculated following the aerodynamic bulk parameterization[37,38]. After masking the seasonal ice-

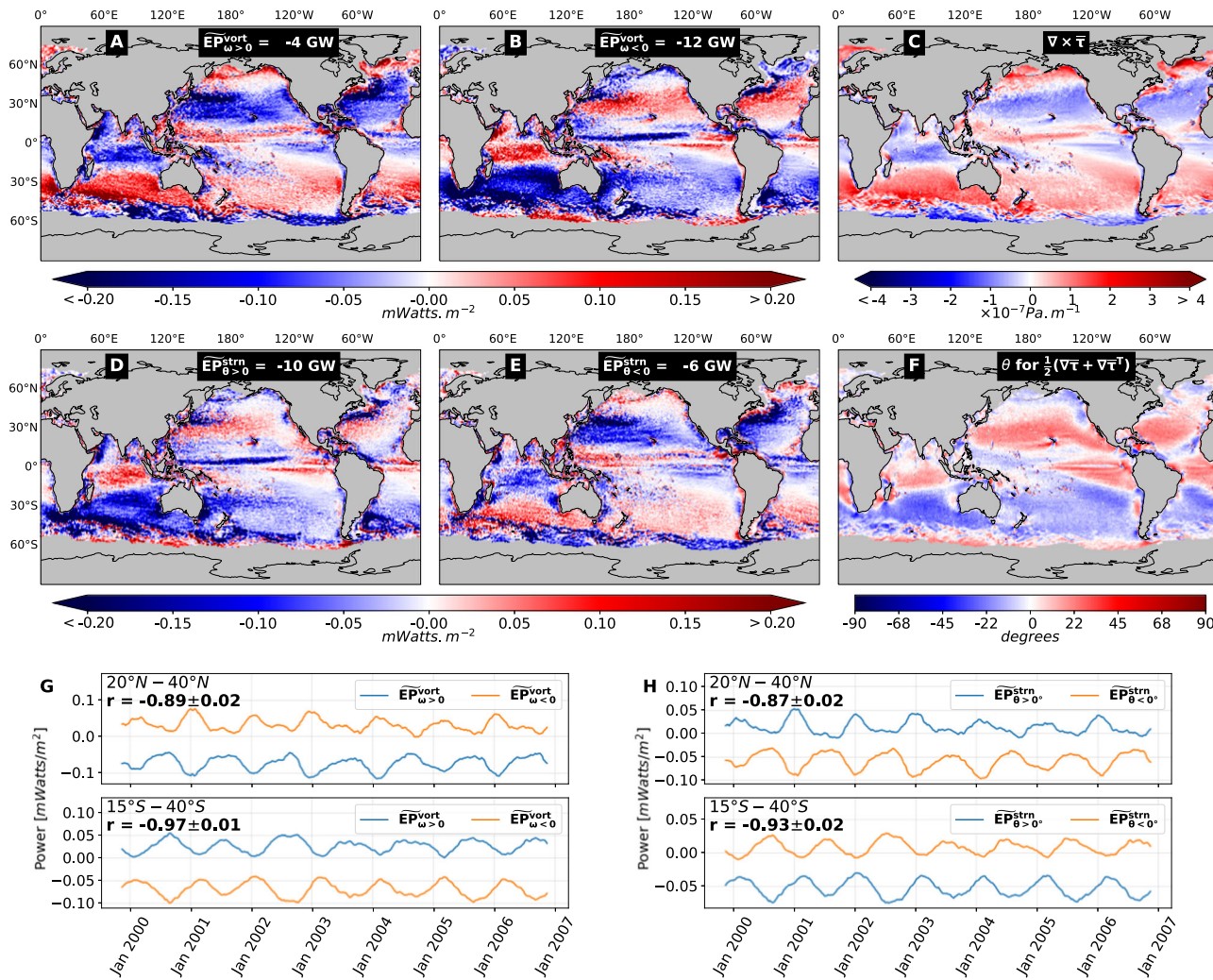

**Fig. 3 | Unraveling inherent asymmetry of energy transfer from winds to ocean weather.** Top/Bottom row shows the inherent asymmetry in wind energization of vortical/straining ocean mesoscale flows. **A**, **B** Wind work on flows with positive/negative (anti/clockwise) vorticity, which are cyclonic/anticyclonic in the northern hemisphere (NH) and anti/cyclonic in the southern hemisphere (SH). In the sub-tropics ([15°–45°]), cyclonic/anti-cyclonic vortices are damped/energized (blue/red) by winds and the reverse occurs in sub-polar regions. While eddy-damping dominates on a global average (Fig. 1A, B), anticyclonic vortical flows are in fact energized due to inherent wind stress gradients (WSGs) in most of the subtropical oceans, except in strong current regions where the eddies are sufficiently strong such that induced WSGs, which always oppose ocean currents, dominate. **C** is a map of the time-mean wind stress curl component of inherent WSGs acting on the ocean's mesoscales. Comparing (**A**, **B**) to (**C**) demonstrates the prevailing winds' imprint (see Fig. 2F) on the ocean's mesoscale vortical flow. Similar to vortical flows,

**D**, **E** show wind energization of straining ocean flows with a positive/negative polarity (**D**/**E**) based on the angle $\theta$ of the local strain's diverging arm (Fig. 2F). **F** The time-mean angle of the straining WSG, which again demonstrates the prevailing winds' imprint (see Fig. 2F) on the ocean's mesoscale straining flow. **G** A (13 weeks running mean) time series of energization of flows with positive/negative vorticity (blue/orange) at latitudes 20°N–40°N and 15°S–40°S, excluding strong current regions where damping by induced WSGs dominates (see "Methods" and ref. 10). There is clear seasonality in (**G**) with the vortical flow's energization/damping peaking during the local winter. **H** Similar to (**G**) but for the straining mesoscale ocean flow, which shows much the same seasonality. In (**G**, **H**), the correlation coefficients $r \leq -0.9$ between blue and orange plots of each sub-panel highlights the strong temporal correlation of asymmetric wind energization/damping of ocean weather with opposite polarity. Coarse-graining in (**A**–**H**) is at scale $\ell = 100$ km.

covered regions and rain-flags, we calculate the weekly average to get the global coverage of wind stress.

Analysis in the Supplementary Information (SI) uses daily averaged surface currents and wind stress from a high-resolution coupled Community Earth System Model (CESM) simulation[39] over years 50 to 56 of the model's output. The model has lateral resolution of $\approx 0.1° \times 0.1°$ for the ocean and $0.25° \times 0.25°$ for the atmosphere. For consistency with analysis of the satellite data, we mask the same seasonal ice-covered regions.

### Scale decomposition

We use the coarse-graining approach[10,40] to study multiscale wind work. For any scale $\ell$ (in meters), we coarse-grain a field $F$ using the

convolution $*$ defined on the surface of a sphere[4,20,41],

$$\overline{F}_\ell(\mathbf{x}) = G_\ell * F = \int_\Omega F(\mathbf{y}) \, G_\ell(\gamma(\mathbf{x}, \mathbf{y})) \, dS(\mathbf{y}). \quad (5)$$

$G_\ell$ is a normalized convolution kernel, $dS(\mathbf{y})$ is the area measure on the sphere, $\Omega$ is the domain, and $\gamma(\mathbf{x}, \mathbf{y})$ is geodesic distance between any two locations $\mathbf{x}$ and $\mathbf{y}$ on the sphere. For any coarse field $\overline{F}_\ell$, the complementary high-pass field containing scales smaller than $\ell$ is

$$F'_\ell = F - \overline{F}_\ell. \quad (6)$$

Following ref. [10], the kernel we use is

$$G_\ell(\gamma) = A\,(0.5 - 0.5\tanh((\gamma - \ell/2)/10.0)), \tag{7}$$

which is essentially a graded Top-Hat kernel with normalizing factor $A$ to ensure $\int dS\, G_\ell = 1$.

**Deriving the wind work proxy**
From Eq. (1),

$$
\begin{aligned}
EP_\ell &= \overline{(\boldsymbol{\tau}\cdot\mathbf{u})}_\ell - \overline{\boldsymbol{\tau}}_\ell\cdot\overline{\mathbf{u}}_\ell \\
&= \overline{((\overline{\boldsymbol{\tau}}_\ell + \boldsymbol{\tau}'_\ell)\cdot(\overline{\mathbf{u}}_\ell + \mathbf{u}'_\ell))}_\ell - \overline{(\overline{\boldsymbol{\tau}}_\ell + \boldsymbol{\tau}'_\ell)}_\ell\cdot\overline{(\overline{\mathbf{u}}_\ell + \mathbf{u}'_\ell)}_\ell \\
&= \overline{(\overline{\boldsymbol{\tau}}_\ell\cdot\overline{\mathbf{u}}_\ell)}_\ell - \overline{(\overline{\boldsymbol{\tau}}_\ell)}_\ell\cdot\overline{(\overline{\mathbf{u}}_\ell)}_\ell + \text{terms with primes}
\end{aligned}
\tag{8}
$$

Equation (8) is exact. If the spectrum of each of the fields $\boldsymbol{\tau}$ and $\mathbf{u}$ decays faster than $k^{-1}$ over the wavenumber range $k \gtrsim \ell^{-1}$, then the terms with primes $(\dots)'$ are subdominant[42–44]. These scaling conditions, which imply that $EP_\ell$ is ultraviolet scale-local at $\ell$[23], do not necessarily require a power-law scaling, only that the spectrum decays sufficiently fast. For example, the conditions are satisfied if the spectrum decays exponentially and are violated near length-scales of the ocean's mesoscale peak ($\ell \approx 300$ km on a global average) or larger, where the spectral scaling becomes too shallow or has a positive slope at $\ell > 300$ km[2]. For the precise technical conditions, see ref. [22], and for a discussion of the physical connection to smoothness along with examples, see ref. [26].

Ultraviolet scale-locality of a multiscale process such as $EP_\ell$ is a fundamental physical property that is closely related to gauge invariance[21,25]. It implies that contributions from length-scales $\delta < \ell$ to $EP_\ell$ decay at least as fast as $(\delta/\ell)^\sigma$ (with an exponent $\sigma > 0$) and that the dominant contribution to $EP_\ell$ comes from length-scales smaller by a multiplicative factor of $O(1)$[23]. For our purposes here, ultraviolet scale-locality justifies neglecting all the $(\dots)'$ terms in Eq. (8),

$$EP_\ell \approx \overline{(\overline{\boldsymbol{\tau}}_\ell\cdot\overline{\mathbf{u}}_\ell)}_\ell - \overline{(\overline{\boldsymbol{\tau}}_\ell)}_\ell\cdot\overline{(\overline{\mathbf{u}}_\ell)}_\ell. \tag{9}$$

From here, $\widetilde{EP}_\ell$ can be derived in a rather straightforward manner. Consider a 2-dimensional domain that is flat, for simplicity. A convolution with a symmetric kernel such as in Eq. (7) can be written as

$$\overline{\mathbf{u}}_\ell(\mathbf{x}) = \int d^2\mathbf{r}\, G_\ell(\mathbf{r} - \mathbf{x})\, \mathbf{u}(\mathbf{r}) = \int d^2\mathbf{r}\, G_\ell(\mathbf{r})\, \mathbf{u}(\mathbf{x} + \mathbf{r}). \tag{10}$$

An equivalent expression can be derived on the surface of a sphere, with spatial translations replaced with rotations[20]. Therefore,

$$\overline{(\overline{\mathbf{u}}_\ell)}_\ell(\mathbf{x}) = \int d^2\mathbf{r}\, G_\ell(\mathbf{r})\, \overline{\mathbf{u}}_\ell(\mathbf{x} + \mathbf{r}) \tag{11}$$

$$\approx \int d^2\mathbf{r}\, G_\ell(\mathbf{r})\left(\overline{\mathbf{u}}_\ell(\mathbf{x}) + \mathbf{r}\cdot\boldsymbol{\nabla}\overline{\mathbf{u}}_\ell(\mathbf{x})\right) \tag{12}$$

$$= \overline{\mathbf{u}}_\ell(\mathbf{x}) + \left(\int d^2\mathbf{r}\, \mathbf{r}\, G_\ell(\mathbf{r})\right)\cdot\boldsymbol{\nabla}\overline{\mathbf{u}}_\ell(\mathbf{x}) \tag{13}$$

$$= \overline{\mathbf{u}}_\ell(\mathbf{x}). \tag{14}$$

Expression (12) follows from a first-order Taylor-series expansion. Note that a Taylor series expansion is not possible (series does not converge) for a field $\mathbf{u}$ in Eq. (10) with a power-law scaling shallower than $k^{-3}$, which precludes most turbulent flows, including geostrophic mesoscales and submesoscales. Otherwise, turbulence could be solved using Taylor series. Rather, in Eq. (11),

we are performing a Taylor expansion of $\overline{\mathbf{u}}_\ell$, which is smooth and is guaranteed to converge for $|\mathbf{r}| < \ell$[22,26]. Equation (13) follows from the kernel being normalized, $\int d^2\mathbf{r}\, G_\ell(\mathbf{r}) = 1$. Equation (14) follows from the symmetric property of the kernel, $\int d^2\mathbf{r}\, \mathbf{r}\, G(\mathbf{r}) = 0$. Similarly,

$$\overline{(\overline{\boldsymbol{\tau}}_\ell)}_\ell(\mathbf{x}) \approx \overline{\boldsymbol{\tau}}_\ell(\mathbf{x}) \tag{15}$$

to leading order. A similar treatment of the first term in Eq. (9) yields (using Einstein summation notation)

$$
\begin{aligned}
\overline{(\overline{\boldsymbol{\tau}}_\ell\cdot\overline{\mathbf{u}}_\ell)}_\ell(\mathbf{x}) &\approx \overline{\boldsymbol{\tau}}_\ell\cdot\overline{\mathbf{u}}_\ell \\
&\quad + \partial_k\overline{(\tau_i)}_\ell\, \partial_j\overline{(u_i)}_\ell \int d^2\mathbf{r}\, r_k r_j\, G_\ell(\mathbf{r})
\end{aligned}
\tag{16}
$$

$$
\begin{aligned}
&= \overline{\boldsymbol{\tau}}_\ell\cdot\overline{\mathbf{u}}_\ell \\
&\quad + \partial_k\overline{(\tau_i)}_\ell\, \partial_j\overline{(u_i)}_\ell\, \frac{\delta_{kj}}{2}\ell^2 \int d^2\mathbf{r}\, \left|\frac{\mathbf{r}}{\ell}\right|^2 G_\ell(\mathbf{r})
\end{aligned}
\tag{17}
$$

$$= \overline{\boldsymbol{\tau}}_\ell\cdot\overline{\mathbf{u}}_\ell + \frac{1}{2}\, M_2\,\ell^2\, \partial_j\overline{(\tau_i)}_\ell\, \partial_j\overline{(u_i)}_\ell. \tag{18}$$

Equation (17) follows from Eq. (16) due to the kernel's symmetry, $\int d^2\mathbf{r}\, \mathbf{r}\, G(\mathbf{r}) = 0$[26]. An equivalent derivation yields the same result on the surface of a sphere. In Eq. (18), the kernel's second moment $M_2 \equiv \int G_\ell(\gamma)\frac{\gamma^2}{\ell^2}\, dS$ depends on the kernel shape and taken to be 0.441 in our analysis at scale $\ell = 100$ km. Finally, combining Eqs. (14), (15), (18) into the key approximation (9) yields the expression for the proxy,

$$
\begin{aligned}
EP_\ell &\approx \overline{(\overline{\boldsymbol{\tau}}_\ell\cdot\overline{\mathbf{u}}_\ell)}_\ell - \overline{(\overline{\boldsymbol{\tau}}_\ell)}_\ell\cdot\overline{(\overline{\mathbf{u}}_\ell)}_\ell \\
&\approx \frac{1}{2}\, M_2\,\ell^2\, \partial_j\overline{(\tau_i)}_\ell\, \partial_j\overline{(u_i)}_\ell \equiv \widetilde{EP}_\ell.
\end{aligned}
\tag{19}
$$

At any scale $\ell$, the approximation improves with steeper spectra because of improved ultraviolet locality. Conversely, the approximation deteriorates as $\ell$ approaches a spectral peak (shallower spectra). This is demonstrated in Fig. S10 in the SI. That $\overline{(\boldsymbol{\tau}'_\ell\cdot\mathbf{u}'_\ell)}_\ell$ in Eq. (8) is subdominant to the term in Eq. (9) highlights a fundamental difference between a lengthscale decomposition and a Reynolds decomposition. In the latter, $\langle\boldsymbol{\tau}'\cdot\mathbf{u}'\rangle$ is the only nonzero term in $\langle\boldsymbol{\tau}\cdot\mathbf{u}\rangle - \langle\boldsymbol{\tau}\rangle\cdot\langle\mathbf{u}\rangle$, which is the Reynolds analogue of Eq. (8) (see refs. [26,45] for further discussion).

The wind work proxy can be decomposed exactly into energy transfer to straining and vortical ocean flows (Eq. (3)),

$$\widetilde{EP}_\ell = \underbrace{\frac{1}{2}M_2\ell^2\overline{T}_{ij}^{strn}\overline{S}_{ij}}_{\widetilde{EP}^{strn}} + \underbrace{\frac{1}{2}M_2\ell^2\overline{T}_{ij}^{vort}\overline{\Omega}_{ij}}_{\widetilde{EP}^{vort}}. \tag{20}$$

Equation (20) follows directly from the decomposition of ocean current gradients into symmetric and skew-symmetric components, $\boldsymbol{\nabla}\overline{\mathbf{u}} = \partial_j\overline{(u_i)} = \overline{S}_{ij} + \overline{\Omega}_{ij}$. Here, $S_{ij} = (\partial_j u_i + \partial_i u_j)/2$ is the strain rate (symmetric) tensor, and $\Omega_{ij} = (\partial_j u_i - \partial_i u_j)/2$ is the rotation rate (skew-symmetric) tensor. The latter is related to vorticity $\boldsymbol{\omega} = \boldsymbol{\nabla}\times\mathbf{u}$ through $\Omega_{ij} = -1/2(\epsilon_{ijk}\omega_k)$, where $\epsilon_{ijk}$ is the Levi-Civita permutation symbol (e.g., ref. [46]). Similar to ocean current gradients, the wind stress gradients can be decomposed into symmetric and skew-symmetric components, $\boldsymbol{\nabla}\overline{\boldsymbol{\tau}} = \partial_j\overline{(\tau_i)} = \overline{T}_{ij}^{strn} + \overline{T}_{ij}^{vort}$. Superscripts 'strn' and 'vort' denote the respective straining and vortical characters of the energy transfer in Eq. (20) (and Eq. (3) in main text), which does not include cross-interactions between symmetric and skew-symmetric components because their tensorial contraction vanishes identically (e.g., ref. [46]).

From Eq. (20) (or Eq. (3) in main text), we can derive Eq. (4) using the wind stress bulk formulation[38],

$$\boldsymbol{\tau} = \rho_{air} C_d |\mathbf{u}_a - \mathbf{u}|(\mathbf{u}_a - \mathbf{u}). \tag{21}$$

Here, $\rho_{air} \approx 1.2$ kg/m³ is air density and $C_d = O(10^{-3})$ is the coefficient of drag. Since wind speed is much larger than ocean current speed, typically by $O(10)$ to $O(100)$, we have $|\mathbf{u}_a - \mathbf{u}_o| \approx |\mathbf{u}_a|$. Moreover, wind speed is dominated by scales $>O(10^3)$ km[47,48], implying a separation of scales between those of wind and ocean velocities and justifies the following approximation of the stress gradient at scales larger than $O(100)$ km (see also Eqs. 24–25 in ref. [49]):

$$\partial_j \overline{\boldsymbol{\tau}} \approx \rho_{air} C_d |\mathbf{u}_a|(\partial_j \overline{\mathbf{u}}_a - \partial_j \overline{\mathbf{u}}). \tag{22}$$

It follows from Eq. (20) (or Eq. (3) in main text) that

$$\widetilde{EP}_\ell \approx \alpha \, |\mathbf{u}_a| \, \ell^2 [\overline{\mathbf{S}}_a : \overline{\mathbf{S}} + \overline{\boldsymbol{\omega}}_a \cdot \overline{\boldsymbol{\omega}} - (|\overline{\mathbf{S}}|^2 + |\overline{\boldsymbol{\omega}}|^2)], \tag{23}$$

where $\alpha = \rho_{air} C_D M_2/2$.

### Defining regions

Following ref. [10], we generate masks for oceanic regions of interest for the plots in Fig. 1E and in Figs. S1, S7–S10. The equatorial region is the ±8° band, and the Southern Ocean region is the [35°–65°S] band. The remaining regions are irregular and are intended to select strongly eddying regions with strong currents. Specifically, the masks satisfy $\frac{1}{2}|\langle \mathbf{u}_o \rangle|^2 + \frac{1}{2}\langle |\mathbf{u}'_o|^2 \rangle > 0.1 \, \text{m}^2/\text{s}^2$ in the Gulf Stream and Kuroshio, and $\frac{1}{2}|\langle \mathbf{u}_o \rangle|^2 + \frac{1}{2}\langle |\mathbf{u}'_o|^2 \rangle > 0.05 \, \text{m}^2/\text{s}^2$ in the remaining regions. Subject to these thresholds, the masks lie within [35°–70°S] (ACC), [15°–85°W, 23°–55°N] (Gulf Stream), [120°–180°E, 23°–50°N] (Kuroshio), [0°–45°E, 15°–40°S] (Agulhas), and [40°–75°W, 35°–60°S] (Brazil-Malvinas).

### Data availability

Level 3 processed QuikSCAT wind measurements spanning the period of October 1999 to December 2006 used to calculate wind stress and can be accessed at https://data.marine.copernicus.eu/product/WIND_GLO_PHY_L3_MY_012_005/files?subdataset=cmems_obs-wind_glo_phy_my_l3-quikscat-seawinds-asc-0.25deg_P1D-i_202311. Geostrophic current data from AVISO Ssalto/Duacs daily sea level anomalies, which is distributed by Copernicus Marine Environment Monitoring Service (CMEMS), is used for ocean currents and can be accessed at https://doi.org/10.48670/moi-00148. Daily averaged wind stress, sea surface height and ocean surface current output variables from Community Earth System Model (CESM) simulation[39] from simulation year 50 to 56 has been used for the plots in Supplementary Materials. The data can be downloaded from https://www.earthsystemgrid.org/dataset/ucar.cgd.asd.hybrid_v5_rel04_BC5_ne120_t12_pop62.ocn.proc.daily_ave.html. The processed data to produce the plots in the main text is available at https://doi.org/10.5281/zenodo.14170158.

### Code availability

The coarse-graining FlowSieve package[50] is publicly available at https://doi.org/10.5281/zenodo.14553091 and the post processing codes to reproduce the figures are available at https://doi.org/10.5281/zenodo.14170158.

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

## Acknowledgements

We thank Hyodae Seo and Susan Wijffels for helpful discussions. This research was funded by US NASA grant 80NSSC18K0772 and US NSF grant OCE-2123496. H.A. was also supported by US DOE grants DE-SC0020229, DE-SC0014318, and DE-SC0019329, US NSF grants PHY-2020249 and PHY-2206380, and US NNSA grants DE-NA0003856, DE-NA0003914, DE-NA0004134. S.R. was also supported by NOAA grant NA22OAR4310598. J.T.F. was supported by NASA grants 80NSSC23K098, 80NSSC19K1256, and 80NSSC21K0713. Computing time was provided by the National Energy Research Scientific Computing Center (NERSC) under Contract No. DE-AC02-05CH11231, NASA's High-End Computing (HEC) Program through the NASA Center for Climate Simulation (NCCS) at Goddard Space Flight Center, and the Texas Advanced Computing Center (TACC) under ACCESS allocation grant EES220052.

## Author contributions

S.R. carried out most of the data analysis. H.A. conceived the initial idea. S.R., J.T.F., and H.A. contributed to interpreting the results and writing of the manuscript.

## Competing interests

The authors declare no competing interests.
