## [Transparent Peer Review file · Nature Communications]

Atmospheric Wind Energization of Ocean Weather

Corresponding Author: Professor Hussein Aluie

Version 0:

Reviewer comments:

Reviewer #1

(Remarks to the Author)

This manuscript deals with mesoscale oceanic eddies and explores, using both satellite and model data, effect of winds on the upper ocean vorticity and strain rates. First, it is shown that wind damps strain and vorticity at an equal rate globally. Second, it is shown that subtropical winds outside strong current regions damp oceanic cyclones and energize anticyclones, while subpolar winds have the opposite effect. A similar pattern emerges for oceanic strain, where subtropical convergent flow is damped along the west-equatorward east-poleward direction and energized along the east-equatorward west-poleward direction.

The idea of wind work proxy that provides distinct energy inputs into the vortical and straining motions forced by the corresponding vortical and straining wind stress components is interesting. However, in the future it would be useful to estimate and assess roles of the neglected cross-terms.

This work is interesting, of good quality and publishable as it is.

Major comment:

My main (and the only) suggestion to the authors should not be taken as a criticism of what they have done. Neither it should be considered as my demand to make the corresponding revision, as this is only friendly suggestion for some future extension, unless they find it interesting to add to the present paper. The idea is to implement a Helmholtz decomposition of the ocean velocity field into purely rotational and divergent components: let's say R-component and D-component. The R-component can be geographically partitioned into its cyclonic and anticyclonic regions, based on the velocity curl sign. (Also, D-component can be partitioned into its divergent and convergent regions, based on the velocity divergence sign.) In this case wind stress energy inputs into each component, and even each region, are not only explicit, exact and straightforward, but will also yield interesting conclusions about wind energy input and roles of mesoscale eddies. Furthermore, it is simple to filter out eddies and wind stress in various ways and, thus, to find contributions from different scales. Finally, the wind stress itself can be Helmholtz-decomposed, and this will allow to identify exactly all 4 energy inputs: R-to-R, R-to-D, D-to-R, and D-to-D (and these can be inspected conditionally, for cyclones/anticyclones, etc).

Minor comment:

[1] A mechanism by which wind affects mesoscale eddies and ultimately reduces energy stored in the large-scale circulation is discussed in a paper by Hogg and co-authors:

Hogg, A., W. Dewar, P. Berloff, S. Kravtsov, and D. Hutchinson, 2009: The effects of mesoscale ocean-atmosphere coupling on the large-scale ocean circulation. *J. Climate*, 22, 4066–4082.

It may be worth mentioning where authors discuss wind/mesoscale effects.

Reviewer #2

(Remarks to the Author)

Version 1:

Reviewer comments:

Reviewer #2

(Remarks to the Author)

Having read both the authors' responses and the paper again, I am very happy with the modifications made in the theory section. This now reads really well and is extremely clear overall. More importantly, the theoretical treatment is now self-contained and does not need a reading of Eyink's paper, which while foundational and elegant work in many ways, is a substantial investment of time even for the theoretically inclined oceanographer. Overall I have really enjoyed reading the paper and have gained quite a few insights from the primary theoretical result, it's derivation and consequences for the air-sea flux.

Reply to Reviewer #1

Manuscript NCOMMS-24-46879-T
“Atmospheric Wind Energization of Ocean Weather”
by S. Rai, J. T. Farrar, and H. Aluie

We thank the reviewer for their time evaluating our manuscript and for recommending its publication. We have made minor revisions (tracked in in green in the manuscript) in response to both reviewers’ comments. These include revising the Methods section to simplify the mathematical derivations.

Below is a point-by-point detailed response (in blue) to the reviewer’s comments (in black).

REPLY to COMMENTS:

This manuscript deals with mesoscale oceanic eddies and explores, using both satellite and model data, effect of winds on the upper ocean vorticity and strain rates. First, it is shown that wind damps strain and vorticity at an equal rate globally. Second, it is shown that subtropical winds outside strong current regions damp oceanic cyclones and energize anticyclones, while subpolar winds have the opposite effect. A similar pattern emerges for oceanic strain, where subtropical convergent flow is damped along the west-equatorward east-poleward direction and energized along the east-equatorward west-poleward direction.

The idea of wind work proxy that provides distinct energy inputs into the vortical and straining motions forced by the corresponding vortical and straining wind stress components is interesting. However, in the future it would be useful to estimate and assess roles of the neglected cross-terms.

This work is interesting, of good quality and publishable as it is.

Major comment:

My main (and the only) suggestion to the authors should not be taken as a criticism of what they have done. Neither it should be considered as my demand to make the corresponding revision, as this is only friendly suggestion for some future extension, unless they find it interesting to add to the present paper. The idea is to implement a Helmholtz decomposition of the ocean velocity field into purely rotational and divergent components: let’s say R-component and D-component. The R-component can be geographically partitioned into its cyclonic and anticyclonic regions, based on the velocity curl sign. (Also, D-component can be partitioned into its divergent and convergent regions, based on the velocity divergence sign.) In this case wind stress energy inputs into each component, and even each region, are not only explicit, exact and straightforward, but will also yield interesting conclusions about wind energy input and roles of mesoscale eddies. Furthermore, it is simple to filter out eddies and wind stress in various ways and, thus, to find contributions from different scales. Finally, the wind stress itself can be Helmholtz-decomposed, and this will allow to identify exactly all 4 energy inputs: R-to-R, R-to-D, D-to-R, and D-to-D (and these can be inspected conditionally, for cyclones/anticyclones, etc).

We thank you for raising this issue, because it is an idea that will occur to many readers. Please note that in the main text, we analyze the geostrophic flow from satellite altimetry. A geostrophic flow is automatically

Helmholtz decomposed because it is divergence-free (or solenoidal). Variations in the Coriolis parameter introduce only small deviations and on scales much larger than 100 km. This underscores that a Helmholtz decomposition does not separate vorticity from strain. This is made precise in the discussion in the Supplementary Information under the first section titled “Strain in Divergence-free Flows”.

We have now added the following text to the introduction:

An occasional misconception is that a Helmholtz decomposition can separate vorticity from strain, with the latter mistakenly regarded as being solely due to the potential flow accounting for divergent motions (see Supplementary Information, SI). In fact, strain is also an essential constituent of divergence-free (or solenoidal) flows, including the oceanic mesoscales in geostrophic balance where strain dominated regions account for approximately half the KE (Figs. S1,S9 in SI)..

Minor comment:

[1] A mechanism by which wind affects mesoscale eddies and ultimately reduces energy stored in the large-scale circulation is discussed in a paper by Hogg and co-authors: Hogg, A., W. Dewar, P. Berloff, S. Kravtsov, and D. Hutchinson, 2009: The effects of mesoscale ocean-atmosphere coupling on the large-scale ocean circulation. *J. Climate*, 22, 4066–4082. It may be worth mentioning where authors discuss wind/mesoscale effects.

Thank you for bringing this paper to our attention. We now mention it in the introductory paragraph.

Reply to Reviewer #2

Manuscript NCOMMS-24-46879-T
“Atmospheric Wind Energization of Ocean Weather”
by S. Rai, J. T. Farrar, and H. Aluie

We thank the reviewer for their time evaluating our manuscript and for recommending its publication. We have made minor revisions (tracked in in green in the manuscript) in response to both reviewers’ comments. These include revising the Methods section to simplify the mathematical derivations.

Below is a point-by-point detailed response (in blue) to the reviewer’s comments (in black).

REPLY to COMMENTS: The current work is a re-examination of the important but frivolously named ‘eddy-killing’ mechanism that results from the relative motion of ocean and atmosphere and its effect on the stress at the air-sea interface. The authors demonstrate in coupled mesoscale-resolving ocean model data that, the relative wind effects not just the vortically dominant regions in the ocean (i.e. eddies) but also the strain dominated regions to a nearly equal degree by magnitude. Central to this study, from which all the results here follow, is the derivation of the approximation for the scale-wise windwork as

$$EP_\ell = \overline{\boldsymbol{\tau} \cdot \mathbf{u}_\ell} - \overline{\boldsymbol{\tau}_\ell \cdot \mathbf{u}_\ell} \approx \ell^2 \nabla \boldsymbol{\tau}_\ell : \nabla \mathbf{u}_\ell \quad (1)$$

While seemingly simple in structure and obscure in origins, this immediately has pretty strong implications because the scalar tensor product can be broken up into corresponding symmetric and antisymmetric tensor products implying that the wind-stress curl in the atmosphere works against the vorticity while the symmetric component of stress gradient (that the authors refer to as WSG) works against strain. The schematic in Fig.2 offers a nice pictorial representation of this equation, basically. The authors test the accuracy of this approximation through a correlation coefficient and it is generally pretty reasonable globally. Next the authors examine the contributions from the strain and vorticity regions and find them to have similar spatial averaged contributions worldwide but with out-of-phase seasonal cycles and different global patterns which generally seem intuitive and well explained.

Overall, I find this work to be a genuinely insightful addition to this area of relative stress at the air-sea interface and its net drag effects on suppressing oceanic mesoscales and applaud the authors in resisting the urge to add more theatrical terminology along the lines of ‘eddy killing’ i.e. ‘strain killing’! The results here have broad relevance in large scale ocean modeling with implications at climatic scales. As such I recommend publication subject to some **Minor comments**; However, I really want the authors to address my comments below.

This is because I like the overall presentation and the results a lot but have concerns about the supplementary information which can be explained in vastly simpler form as below which should, I think, simplify and help improve the manuscript especially in a broad impact journal:

- I found the approximate derivation of EP_ℓ to be overdone and unnecessary, hiding the fundamental approximations being done here. I understand this closely follows Eyink 2006 but that is no excuse for a simpler and, something that took me a great deal of effort after reading Eyink 2006, actually obvious treatment; especially given that this is in a broad impact journal. The authors can keep all the ultraviolet-scale locality arguments, but the rest can be vastly condensed and simplified.

Here is a suggestion to simplify this or at least help clarify the treatment and help a lot more oceanographers, who are not somewhat as theoretically inclined as I might be. Basically this is what the authors are doing as far as I can tell. First write

$$F(\mathbf{x}) = \overline{F}_\ell(\mathbf{x}) + F'_\ell(\mathbf{x}) \quad (2)$$

Next, completely ignore all the wavelet-style multi-scale expansion stuff. So you do this for both $\boldsymbol{\tau}$ and \mathbf{u} . Then substitute the above in the wind work as (I am abusing notation and the (\cdot) is basically the tensor contraction).

$$EP_\ell = \overline{\boldsymbol{\tau} \cdot \mathbf{u}_\ell} - \overline{\boldsymbol{\tau}_\ell} \cdot \overline{\mathbf{u}_\ell} \quad (3)$$

$$= \overline{(\boldsymbol{\tau}_\ell \cdot \mathbf{u}_\ell)_\ell} - \overline{(\boldsymbol{\tau}_\ell)_\ell} \cdot \overline{(\mathbf{u}_\ell)_\ell} \quad (4)$$

$$+ 2 \text{ cross terms} + \text{small scale wind work term} \quad (5)$$

$$\approx \overline{(\boldsymbol{\tau}_\ell \cdot \mathbf{u}_\ell)_\ell} - \overline{(\boldsymbol{\tau}_\ell)_\ell} \cdot \overline{(\mathbf{u}_\ell)_\ell} \quad \mathbf{What!?!} \quad (6)$$

i.e. the scale-wise wind work, EP_ℓ can be computed by first filtering the respective stress and velocity field and then computing the scale-wise wind work (i.e. \widetilde{EP}_ℓ)!! I don't know about the authors, but I find the result both highly non-trivial and surprising and would not have believed it a priori without the clear demonstration by the authors in the model. The total wind work (not-scale wise) is dominated by the largest scales (after all for all the effective 'eddy-strain-killing' drag at mesoscales, wind does drive the ocean due to the large scale input) but this result shows that this is true for each scale, i.e. the wind power input within each scale range $[0, \ell]$ are dominated by the by the largest scales at scale ℓ . I think this is a pretty important result to present without inadvertently obfuscating it behind the "multi-scale expansion" and the ultraviolet-scale locality (a discussion of which should appear after arguments above).

The fact that it took me, a moderately theoretically-inclined GFD person, so long to arrive here should speak somewhat to a need in simplifying of the presentation style as suggested above. Next, Eyink's use of 'increments', $F(\mathbf{x}; \mathbf{r}) = F(\mathbf{x} + \mathbf{r}) - F(\mathbf{x})$, while very elegant, are again, completely unnecessary here and add to needless complexity and mystery. In fact, almost all the meat is in Eq 11 (i.e. the approximation above). At this point, you can mindlessly expand each of the terms $\overline{\boldsymbol{\tau}_\ell}$ and $\overline{\mathbf{u}_\ell}$ in Taylor expansions around \mathbf{x} in the integrand, multiply everything out, use the symmetry property of $G(\mathbf{r})$ to set the $\int \mathbf{r}G(\mathbf{r})d\mathbf{r} = 0$, take the lowest order term and you get Eq 18a with no extra conceptual additions. Basically, unless I am missing something critical, Eqs 12 to 16 are pretty unnecessary and 17 can be replaced by the Taylor expansion of \overline{F}_ℓ around \mathbf{x} . (I am also surprised that applied mathematicians of the caliber of Peter Constantin and Edriss Titi have such a trivial identity to their name; apologies for my insolence!) Basically the tradeoff is in maximizing clarity vs elegance in a physics context; as a rule I always prefer the former.

We are very grateful to the reviewer for their effort and time going through the derivations and also for prodding us in this direction. We agree that a simplified (and hopefully more transparent) derivation would be welcome by many of the readers. We have revised the Methods Section accordingly.

- The result in Fig 2 A and B is very nice but it would be helpful to add the total EKE in the strain dominated regions vs eddying regions - this supports the idea that both have similar order of KE reservoir and therefore similar damping mechanism leads to a similar net loss.

We added the requested panels to Figs. S1 and S9. We also added a sentence in the introduction mentioning that "... strain dominated regions account for approximately half the KE (Figs. S1,S9 in SI)".

- lines 5-7 - Strain does not govern frontogenesis in the ocean; it is merely one of the few mechanisms , boundary layer turbulence being another (the so called TTW mechanism).

We were also uneasy with that phrasing. We changed it to, "Strain contributes to... frontogenesis...".

- In Fig. 1 and beyond, the authors use correlation coefficient as a metric but I prefer the coefficient of determination (basically a normalized mean squared error between the truth and model). This paints a more accurate measure of how close mismatch is. This is because the seasonal cycle can introduce a high correlation even in less than accurate models. This metric will bring out the mismatch in the Southern Ocean better relative to the Gulf Stream (Fig 1 E).

Thank you for the suggestion. We now include the value of the coefficient of determination, R^2 , in panels [B], [D] and [E] of our figures Fig. 1 and Fig. S4

- The overall figure aesthetics and clarity are commendable but in Fig 3 G and H, the lat lon ranges can be moved a little closer to the center and without intersecting the top line.

Thank you for this suggestion. We have made changes in panel [G], [H] of Fig 3, S3, S6 and S8.

- Fig 3 F - Why is this the only one with non-open range colorbars?

This is because the angle of symmetry for strain is 90°

Referee’s Report for **Atmospheric Wind
Energization of Ocean Weather** by *Rai et al*
2024

The current work is a re-examination of the important but frivolously named ‘eddy-killing’ mechanism that results from the relative motion of ocean and atmosphere and its effect on the stress at the air-sea interface. The authors demonstrate in coupled mesoscale-resolving ocean model data that, the relative wind effects not just the vortically dominant regions in the ocean (i.e. eddies) but also the strain dominated regions to a nearly equal degree by magnitude. Central to this study, from which all the results here follow, is the derivation of the approximation for the scale-wise windwork as

$$EP_\ell = \overline{\boldsymbol{\tau} \cdot \mathbf{u}_\ell} - \overline{\boldsymbol{\tau}_\ell} \cdot \overline{\mathbf{u}_\ell} \approx \alpha \ell^2 \nabla \overline{\boldsymbol{\tau}_\ell} \cdot \nabla \overline{\mathbf{u}_\ell} \quad (1)$$

While seemingly simple in structure and obscure in origins, this immediately has pretty strong implications because the scalar tensor product can be broken up into corresponding symmetric and antisymmetric tensor products implying that the wind-stress curl in the atmosphere works against the vorticity while the symmetric component of stress gradient (that the authors refer to as WSG) works against strain. The schematic in Fig.2 offers a nice pictorial representation of this equation, basically. The authors test the accuracy of this approximation through a correlation coefficient and it is generally pretty reasonable globally. Next the authors examine the contributions from the strain and vorticity regions and find them to have similar spatial averaged contributions worldwide but with out-of-phase seasonal cycles and different global patterns which generally seem intuitive and well explained.

Overall, I find this work to be a genuinely insightful addition to this area of relative stress at the air-sea interface and its net drag effects on suppressing oceanic mesoscales and applaud the authors in resisting the urge to add more theatrical terminology along the lines of ‘eddy killing’ i.e. ‘strain killing’! The results here have broad relevance in large scale ocean modeling with implications at climatic scales. As such I recommend publication subject to some **Minor comments**; However, I really want the authors to address my comments below.

This is because I like the overall presentation and the results a lot but have concerns about the supplementary information which can be explained in vastly simpler form as below which should, I think, simplify and help improve the manuscript especially in a broad impact journal:

- I found the approximate derivation of EP_ℓ to be overdone and unnecessary, hiding the fundamental approximations being done here. I understand this closely follows Eyink 2006 but that is no excuse for a simpler and, something that took me a great deal of effort after reading Eyink 2006, actually obvious treatment; especially given that this is in a broad impact journal. The authors can keep all the ultraviolet-scale locality arguments, but the rest can be vastly condensed and simplified.

Here is a suggestion to simplify this or at least help clarify the treatment and help a lot more oceanographers, who are not somewhat as theoretically inclined as I might be. Basically this is what the authors are doing as far as I can tell. First write

$$F(\mathbf{x}) = \bar{F}_{ell}(\mathbf{x}) + F'(\mathbf{x})_\ell \quad (2)$$

Next, completely ignore all the wavelet-style multi-scale expansion stuff. So you do this for both $\boldsymbol{\tau}$ and \mathbf{u} . Then substitute the above in the wind work as (I am abusing notation and the (\cdot) is basically the tensor contraction).

$$EP_\ell = \overline{\boldsymbol{\tau} \cdot \mathbf{u}_\ell} - \bar{\boldsymbol{\tau}}_\ell \cdot \bar{\mathbf{u}}_\ell \quad (3)$$

$$= \overline{(\boldsymbol{\tau}_\ell \cdot \mathbf{u}_\ell)_\ell} - \overline{(\boldsymbol{\tau}_\ell)_\ell} \cdot \overline{(\mathbf{u}_\ell)_\ell} \quad (4)$$

$$+ 2 \text{ cross terms} + \text{small scale wind work term} \quad (5)$$

$$\approx \overline{(\boldsymbol{\tau}_\ell \cdot \mathbf{u}_\ell)_\ell} - \overline{(\boldsymbol{\tau}_\ell)_\ell} \cdot \overline{(\mathbf{u}_\ell)_\ell} \quad \mathbf{What!?!} \quad (6)$$

i.e. the scale-wise wind work, EP_ℓ can be computed by first filtering the respective stress and velocity field and then computing the scale-wise wind work (i.e. \tilde{EP}_ℓ)!! I don't know about the authors, but I find the result both highly non-trivial and surprising and would not have believed it *a priori* without the clear demonstration by the authors in the model. The total wind work (not-scale wise) is dominated by the largest scales (after all for all the effective 'eddy-strain-killing' drag at mesoscales, wind does drive the ocean due to the large scale input) but this result shows that this is true for each scale, i.e. the wind power input within each scale range $[0, \ell]$ are dominated by the by the largest scales at scale ℓ . I think this is a pretty important result to present without inadvertently obfuscating it behind the "multi-scale expansion" and the ultraviolet-scale locality (a discussion of which should appear after arguments above). .

The fact that it took me, a moderately theoretically-inclined GFD person, so long to arrive here should speak somewhat to a need in simplifying of the presentation style as suggested above.

Next, Eyink's use of 'increments', $F(\mathbf{x}; \mathbf{r}) = F(\mathbf{x} + \mathbf{r}) - F(\mathbf{x})$, while very elegant, are again, completely unnecessary here and add to needless complexity and mystery. In fact, almost all the meat is in Eq 11 (i.e. the approximation above). At this point, you can mindlessly expand each of the terms $\bar{\tau}_\ell$ and $\bar{\mathbf{u}}_\ell$ in Taylor expansions around \mathbf{x} in the integrand, multiply everything out, use the symmetry property of $G(\mathbf{r})$ to set the $\int \mathbf{r}G(\mathbf{r})d\mathbf{r} = 0$, take the lowest order term and you get Eq 18a with no extra conceptual additions. Basically, unless I am missing something critical, Eqs 12 to 16 are pretty unnecessary and 17 can be replaced by the Taylor expansion of \bar{F}_ℓ around \mathbf{x} . (I am also surprised that applied mathematicians of the caliber of Peter Constantin and Edriss Titi have such a trivial identity to their name; apologies for my insolence!)

Basically the tradeoff is in maximizing clarity vs elegance in a physics context; as a rule I always prefer the former.

- The result in Fig 2 A and B is very nice but it would be helpful to add the total EKE in the strain dominated regions vs eddying regions - this supports the idea that both have similar order of KE reservoir and therefore similar damping mechanism leads to a similar net loss.
- lines 5-7 - Strain does not govern frontogenesis in the ocean; it is merely one of the few mechanisms, boundary layer turbulence being another (the so called TTW mechanism).
- In Fig. 1 and beyond, the authors use correlation coefficient as a metric but I prefer the coefficient of determination (basically a normalized mean squared error between the truth and model). This paints a more accurate measure of how close mismatch is. This is because the seasonal cycle can introduce a high correlation even in less than accurate models. This metric will bring out the mismatch in the Southern Ocean better relative to the Gulf Stream (Fig 1 E).
- The overall figure aesthetics and clarity are commendable but in Fig 3 G and H, the lat lon ranges can be moved a little closer to the center and without intersecting the top line.
- Fig 3 F - Why is this the only one with non-open range colorbars?